# Essential Amino Acids-Rich Diet Decreased Adipose Tissue Storage in Adult Mice: A Preliminary Histopathological Study

**DOI:** 10.3390/nu14142915

**Published:** 2022-07-15

**Authors:** Giovanni Corsetti, Claudia Romano, Silvia Codenotti, Evasio Pasini, Alessandro Fanzani, Francesco S. Dioguardi

**Affiliations:** 1Division of Human Anatomy and Physiopathology, Department of Clinical and Experimental Sciences, University of Brescia, 25023 Brescia, Italy; cla300482@gmail.com; 2Division of Biochemistry, Department of Molecular and Translational Medicine, University of Brescia, 25023 Brescia, Italy; silvia.codenotti@unibs.it (S.C.); alessandro.fanzani@unibs.it (A.F.); 3Italian Association of Functional Medicine, 20855 Lesmo, Italy; evpasini@gmail.com; 4Department of Internal Medicine, University of Cagliari, 9128 Cagliari, Italy; fsdioguardi@gmail.com

**Keywords:** adipose tissue, amino acids, diet, nutrition, immunohistochemistry, mouse

## Abstract

Background: Excess body adipose tissue accumulation is a common and growing health problem caused by an unbalanced diet and/or junk food. Although the effects of dietary fat and glucose on lipid metabolism regulation are well known, those of essential amino acids (EAAs) have been poorly investigated. Our aim was to study the influence of a special diet containing all EAAs on retroperitoneal white adipose tissue (rpWAT) and interscapular brown adipose tissue (BAT) of mice. Methods: Two groups of male Balb/C mice were used. The first was fed with a standard diet. The second was fed with an EAAs-rich diet (EAARD). After 3 weeks, rpWAT and BAT were removed and prepared for subsequent immunohistochemical analysis. Results: EAARD, although consumed significantly less, moderately reduced body weight and BAT, but caused a massive reduction in rpWAT. Conversely, the triceps muscle increased in mass. In rpWAT, the size of adipocytes was very small, with increases in leptin, adiponectin and IL-6 immunostaining. In BAT, there was a reduction in lipid droplet size and a simultaneous increase in UCP-1 and SIRT-3. Conclusions: A diet containing a balanced mixture of free EAA may modulate body adiposity in mice, promoting increased thermogenesis.

## 1. Introduction

Overweight, obesity and related disorders are all conditions on the rise for all age groups in many parts of the world, causing serious social and health problems. The main cause of these conditions is diet, in particular the excessive and/or strongly imbalanced intake of macronutrients such as carbohydrates, lipids and proteins. Although the effects of dietary fat and glucose on lipid metabolism regulation are well known, the effects of amino acids (AAs) have been poorly investigated.

Food intake is crucial in providing energy to fuel body metabolism and everyday activities for all living organisms [1]. Among macronutrients, proteins do not have an inactive compound to draw from within the body, and therefore dietary AAs need to be incorporated into functional proteins. When in need (such as in fasting), structural proteins (especially muscle and skin proteins) function as a reserve of AAs.

AAs are totipotent molecules. They may be transformed not only in proteins but also in carbohydrates and lipids, subsequently acting as fuel molecules. AAs are fundamental for life because they are the precursors of protein synthesis (structural, circulating and enzymes), the main source of nitrogen for mammal metabolism. Unfortunately, approximately half of AAs in proteins cannot be synthesized in metazoans as the genes for their synthesis were lost early in evolution. These AAs are called essential (EAAs) and they must be taken in sufficient quantities through the ingestion of food. Among all EAAs, Leu, Ile and Val have been identified as branched-chain AAs (BCAAs), as they have non-linear aliphatic side chains. BCAAs play a primary role in many metabolic processes.

Other AAs can be synthesized independent of the body, and therefore are defined as non-essential (NEAAs). To survive, all organisms must maintain a full and balanced complement of AAs, especially so if with an excess of EAAs [2,3]. Dietary animal and vegetable proteins are the only source of EAAs, but the exact amount of EAAs introduced daily with the diet is very difficult to establish. Unfortunately, dietary proteins contain an excess of NEAAs (EAA/NEAA ≤ 0.9 at best, with a mean % of approximately 45/55, respectively) [4]. This means that we have to introduce a very large amount of NEAAs to meet the needs of EAAs. Consequently, excess NEAAs must be eliminated through complex metabolic pathways [5]. Furthermore, the amount of EAAs provided in the diet is not always adequate to match animal needs even when they are in good health [6].

In previous work, we showed that aged mice on a standard growth diet, supplemented with a special balanced diet rich in EAAs, showed increased lifespan and improved mitochondrial biogenesis. We also showed that this was linked to morphological and molecular changes in the heart, skeletal muscle and adipose tissue with a reduction in body weight [7,8,9,10]. The protective role of an EAAs-rich diet was also observed in the liver of rats undergoing chronic ethanol consumption [11], in the kidneys of rosuvastatin-treated mice [12] and in wound healing [13]. Furthermore, in vitro data showed that variations in the EAA/NEAA ratio regulate cancer cell survival or death [14]. Globally, these data would suggest that different ratios of EAA/NEAA in the diet affect whole-body metabolism.

Previous studies have also shown that BCAAs play important roles in lipid metabolism regulation [15,16,17]. It has been demonstrated that mice fed with a diet deficient in any BCAA decrease abdominal fat mass and improve insulin sensitivity [18,19,20,21]. In addition, the dietary deficiency of other EAAs for 7 days also significantly reduced abdominal fat mass, possibly caused by increased energy expenditure [22]. Adequate caloric intake is crucial for growth and lifespan. Notably, sufficient caloric intake with insufficient nitrogen supply could cause ‘under-nutrition’ conditions, often synonymous with protein–energy malnutrition [23].

However, quantitative protein under-nutrition is not the only factor in malnutrition. Protein quality also plays a pivotal role. Indeed, qualitative protein malnutrition, due to both social phenomena and junk foods, is largely responsible for metabolic alterations causing an increase in obesity and related chronic diseases. For this reason, nutritional strategies aiming to maintain optimal metabolism and regulate lipid accumulation are of considerable importance.

Adipose tissues (AT), in particular white ones (WAT), are not only an energy storage depot, but also an endocrine organ. The discovery of the endocrinal property of WAT and the regulatory effects of adipocyte-derived hormones on energy metabolism has clearly demonstrated that WAT plays an active role in maintaining energy homeostasis [24,25,26,27]. Another type of AT is the brown (BAT) variant. Unlike WAT, BAT contains numerous smaller lipid droplets, a lot of mitochondria and capillaries network and is highly specialized for this non-shivering thermogenesis [28].

Experimental models have shown that the absence of each EAA causes a reduction in AT [22]. However, this condition is not common in nature, in which the different alimentary proteins possess a mixture of EAAs and, considerably more, NEAAs. Furthermore, the food variety only barely allows the constant absence of one or more EAAs. As a result, the aim of this study was to determine the influence of a balanced special diet with an excess of all free EAAs on retroperitoneal WAT (rpWAT), interscapular BAT and body composition of adult mice.

## 2. Materials and Methods

The experimental protocol was approved and conducted in accordance with the Italian Ministry of Health and complied with the ‘The National Animal Protection Guidelines’. The Ethical Committee for animal experiments of the University of Brescia and the Italian Ministry of Health approved the procedures.

*Diets and Mice.* The EAAs-rich diet (EAARD) contained a mixture of EAAs (84%) and NEAAs (16%) as previously described [2]. This diet was iso-nutrient, iso-caloric and iso-nitrogenous, and was expressly prepared by Dottori Piccioni (Milan, Italy) in accordance with AIN76-A/NIH-7 rules. A standard laboratory rodent food substance (StD) (Mucedola srl, Milan, Italy) was used as the reference diet with a nitrogen source represented by unspecified vegetal and animal (fish) proteins. The composition of the diet is summarized in Table 1.

Twenty adult male Balb/C mice, six-months-old, were randomly assigned to the two groups. The first group (*n* = 8) was fed with StD ad libitum. The second group (*n* = 12) was fed with EAARD ad libitum. The animals were placed in a quiet, temperature and humidity-controlled room, and were kept on a 12/12 h light/dark cycle (lights on from 7 a.m. to 7 p.m.) and were inspected daily by experienced veterinarians. Every three days, body weight (b.w.), food and water consumption were measured. All animals fed with EAARD were deemed ‘not suffering’ according to veterinary advice. No abnormal behavior (such as kyphotic posture and/or lethargy) was observed in animals fed EAARD. The animals maintained their characteristic vitality and strength in supporting themselves when hanging from the cage.

*Samples collections.* At the end of the treatment, the animals were euthanized by cervical dislocation. rpWAT and BAT were immediately removed, quickly washed in physiological solution, fixed overnight in 10% neutral buffered formalin, processed and embedded in paraffin. The 5 μm sections were dewaxed, rehydrated and used for immunohistochemistry (IHC). Other AT samples were fixed with 3% glutaraldehyde in a cacodylate buffer (pH 7.4, 0.2 M) and processed with standard procedures for embedding in Araldite (Sigma-Aldrich Chemical Co., Milan, Italy). Thick sections (approximately 1 µm) were stained with toluidine blue and used for morphometry.

*Immunohistochemistry (IHC).* AT sections were incubated overnight with primary polyclonal anti-ATGL (F7)-HRP (sc-365278-HRP) (WAT and BAT), anti-FAS (G11) (sc: 48357) (WAT and BAT), anti-FADD (G4) (sc-271748) (WAT/BAT), anti-SIRT3 (H-40): (sc-99143) (BAT), and anti-IL6 (M-19)-R (sc-1265) (WAT/BAT)—all from Santa Cruz Biotechnology Inc. (Dallas, Texas). Anti-IL10 (AA 135-178) was from Antibodies-online.com (WAT/BAT). Anti-adiponectin [19F1] (GTX80683) (WAT/BAT), anti-leptin (GTX109204) (WAT/BAT), and anti-UCP1 [GT1345] (GTX632186) (BAT) were all from Gene-Tex (Irvine, CA, USA). All polyclonal antibodies were diluted 1:100 with PBS and monoclonal antibodies were diluted 1:250 with PBS. The sections were processed according to the manufacturer’s protocol and visualized with a rabbit ABC-peroxidase staining system kit (Santa Cruz Biotechnology Inc., Dallas, TX, USA).

In order to exclude incorrect interpretation of immunostaining due to endogenous biotin, we also carried out experiments using the peroxidase–anti-peroxidase detection system. We obtained similar results with both methods. Each set of experiments was performed in triplicate, with each replicate carried out under identical experimental conditions. The IHC control was performed by omitting the primary antibody in the presence of isotype-matched IgGs. The staining intensity in both histochemical and IHC slides was evaluated using an optical Olympus BX50 microscope equipped with an image analysis program (Image Pro-Plus 4.5.1, Immagini e Computer, Milano, Italy) and analyzed quantitatively. The IOD was calculated for arbitrary areas, by measuring 10 fields for each sample using a 20× lens.

*Statistics.* Data are expressed as the mean ± sd. To compare the results of experimental groups, statistical analysis was performed by two-sample unpaired *t*-test (www.meta-calculator.com: accessed on 15 March 2022), and a value of *p* < 0.05 was considered statistically significant.

## 3. Results

EAARD was capable of inducing generalized body changes. Indeed, at the end of treatment (21 days), the b.w. of these mice decreased by approximately 10% compared to StD-fed ones (Figure 1A,B).

The mean daily food intake of EAARD-fed mice was significantly lower, approximately −37% (2.8 ± 0.23 g/day), compared with those of StD-fed animals (4.43 ± 0.33 g/day). The mean daily caloric intake changes proportionally (Table 2). Nevertheless, EAARD-fed animals showed an increase in triceps muscle (approximately +28.6%). Organ weight and normalized organ weight according to diets are listed in Table 3 and Table 4, respectively.

### 3.1. rpWAT

EAARD-fed mice decreased rpWAT mass by approximately 93.7%. The mean size of adipocytes was very small (<500 µm^2^) when compared to the normal size of StD-fed animals (>>500 µm^2^) (Figure 2). We observed significant increases in leptin (Figure 3), adiponectin (Figure 4), IL-6 (Figure 5), and IL-10 (Figure 6) immunostaining density. Furthermore, we observed a pattern, even if non-significant, of fatty acid synthase (FAS) to decrease (Figure 7), FAS-associated protein with death domain (FADD) (Figure 8) and, of peculiar interest, highly significant (*p* < 0.01) adipose triglyceride lipase (ATGL) immunostaining (Figure 9).

### 3.2. BAT

Animals fed with EAARD showed a decrease in BAT mass by approximately 26.6% (Table 3). The adipocyte area was smaller than in StD-fed mice (Figure 10) and lipid droplet number and size were both smaller in EAARD-fed mice. No differences between StD and EAARD were observed in leptin (Figure 3A), adiponectin (Figure 4A), IL-6 (Figure 5A) and FAS staining (Figure 7A), whereas a non-significant pattern of IL-10 expression increase was observed after EAARD treatment (Figure 6A,D–E). In contrast, the uncoupling protein 1 (UCP-1) and SIRT-3 increases in BAT from EAARD mice were more significant (Figure 11 and Figure 12, respectively).

## 4. Discussion

The main result of this study is that animals fed with EAARD had a dramatic reduction in rpWAT volume, which was less relevant in BAT, but had an increase in muscle mass. Those modifications in rpWAT volume were paralleled with a marked reduction in food consumption, leading to just a 10% decrease in animals b.w.

EAARD was consumed in significantly lower quantities (−37%) than StD, thus leading to decreased caloric intake. Noticeably, no proportional reductions in b.w. were observed. This would suggest that the limited caloric intake might not be the only condition accounting for b.w. loss when considering the quality of AAs ingested. It is possible that the low consumption of free EAAs-rich food is sufficient to induce satiety and to provide a metabolically active nitrogen source for the animals’ life. This is in line with previous findings, which reported the deleterious effect of EAA deficiency [2,3]. Our data are in line with previous studies showing that elevated plasma concentrations of EAAs can trigger satiety signals, thereby decreasing food intake [29]. Indeed, in these animals’ rpWAT, we found a significant increase in leptin, IL6 and adiponectin expression.

Leptin is a hormone predominantly produced by adipocytes and enterocytes in the small intestine that helps to regulate energy balance diminishing the sense of hunger and, as a consequence, increasing energy expenditure and diminishing fat storage. The leptin acts on cell receptors of the hypothalamus (arcuate and ventromedial nuclei), as well as other parts of the dopaminergic neurons of the ventral tegmental area, consequently mediating feeding [30,31]. In addition to alterations in insulin, changes in circulating levels of glucose [32], AAs and lipids may also contribute to increases in leptin production in response to meals [33].

In a physiological context, it has been shown that increases in circulating BCAAs contribute to meal-induced increases in leptin. Particularly, the administration of leucine, elicits the increase in leptin after 3h via activation of mTOR [33,34]. Furthermore, it has been demonstrated that inhibitors of PI3-kinase, AKT, or mTOR block insulin-stimulated leptin biosynthesis [35]. On the contrary, fasting, with a decline in energy or nutrient availability, decreased mTOR and increased AMPK signaling, contributing to decreased leptin levels [33]. Increases in leptin expression and secretion from adipocytes have been linked to insulin stimulation [36], cell glucose uptake [37] and the availability of energy substrates [38], all of which are indicative of an anabolic state. The special diet used in this study contained all free EAAs in stoichiometric ratio with the predominance of BCAAs [5] given that previous works have demonstrated that this mixture stimulates mTOR activation [9]. This factor may explain the persistence of high leptin expression observed in rpWAT.

Simultaneous with the increase in leptin due to EAARD, we observed an increase in IL-6. This agrees with previous research showing that IL-6 acts on AT to increase leptin secretion suppressing hunger and increasing lipolysis [39]. IL-6 is a potent cytokine that acts at key sites of metabolic regulation in many tissues. In healthy humans IL-6 increased insulin-stimulated glucose disposal, increased lipolysis, glucose, fatty acid oxidation, and energy expenditure [40,41,42]. Experimental studies in animal models indicate that IL-6 acts on the central nervous system, inducing the suppression of food intake and body weight by stimulating the glucagon-like peptide-1 (GLP-1) receptor [43]. Mice over-expressing IL-6 and the soluble IL-6R are significantly smaller and have a large reduction in body weight when compared with WT mice, which was mainly triggered by loss of adiposity [44]. The effect of these changes would limit energy deposition within WAT by direct effects, as well as possible negative feedback of food intake caused by the leptin [45].

The effects of IL-6 on metabolism need signal integration between many cell types [46]. Although IL-6 is primarily considered a pro-inflammatory cytokine, it is also involved in both anti-inflammatory and non-inflammatory mechanisms [47]. In the absence of inflammation, between 10 and 35% of circulating IL-6 may derive from AT [48].

The alteration of lipid homeostasis underlies the etiology of some common chronic diseases such as obesity, cardiovascular and liver diseases, and metabolic diseases are also associated with chronic inflammation [49]. Previously, it has been demonstrated that dietary supplementation with a special EAAs mixture prevents oxidative damage in the heart and muscles [9], counteracts metabolic and functional damage in heart of diabetic rat [50], reduces liver damage induced by ethanol consumption [11] and increases lifespan [3,9], suggesting that a diet with excess EAAs promotes and maintains an anti-inflammatory mechanism. This factor is also confirmed by a slight increase in IL-10 in both rpWAT and BAT from EAARD-fed mice. Indeed, the anti-inflammatory role of IL-6 is mediated through its inhibitory effects on TNFα and IL-1 and its activation of IL-10. As an anti-inflammatory cytokine, IL-10 has been shown to have a protective role on the formation and stability of atherosclerotic lesions [51] in rodents. Humans with low IL-10 serum levels had an increased risk of stroke and type 2 diabetes and are associated with the metabolic syndrome [52,53,54]. Furthermore, it has been found that adiponectin upregulated anti-inflammatory IL-10 secretion and tissue inhibitors of metalloproteinase-1 in human macrophages [55]. In addition, the protective role of EAARD in rpWAT is supported by the tendency of FADD to reduce. FADD is an adaptor protein that plays a pivotal role as controller of many essential cellular processes, in particular apoptosis [56,57]. The cytoplasmic decreases that have been observed would suggest its anti-apoptotic effect.

Adiponectin is another adipocyte-derived hormone that actively regulates energy homeostasis [58]. Adiponectin circulates as a multimer, activating lipid metabolism in target tissues. This is involved in regulating glucose levels as well as fatty acid breakdown, so the prominent function is to improve insulin sensitivity [59] and, unlike other adipocyte-derived hormones, its gene expression and blood concentrations are inversely associated with body-mass index [60]. Decrease in circulating adiponectin in the prediabetic state precedes the development of insulin resistance [61]. Therefore, hypoadiponectinemia has been considered to be an underlying mechanism of insulin resistance in obesity and type 2 diabetes [27,62,63]. By increasing the concentration of adiponectin, the anabolism of fat is blocked, and the liver inhibits the synthesis of fatty acids and cholesterol. However, changes in metabolism of energy are activated; the oxidation of fatty acids and the storage of glucose in the cells is promoted, decreasing the production of hepatic glucose. Adiponectin also acts on the hypothalamic centers reducing the sense of hunger. Indeed, transgenic mice with increased adiponectin have reduced adipocyte differentiation and increased energy expenditure associated with mitochondrial uncoupling [64]. Furthermore, adiponectin in combination with leptin has been shown to completely reverse insulin resistance in mice [65]. The strong increases in adiponectin observed in rpWAT of EAARD-fed mice suggest the pivotal role of the EAAs mixture in the control of energy metabolism and lipid catabolism.

ATGL is the enzyme responsible for the initial step of triglycerides dismantling and so of lipid beta-oxidation, influencing the whole-body energy homeostasis [66]. ATGL protein is highly expressed in AT but, at more reduced levels, also in several non-AT also [67]. In adipocytes, ATGL protein levels increase upon fasting [68], and thus control whole-body FA and energy supply dependent on the nutritional status. On the contrary, loss of ATGL in adipocytes causes a faster sympathetic activation of AT to allow FA mobilization [69]. Our results show a decrease in ATGL immunostaining in WAT of EAARD-fed mice, so there is a possibility that this decrease could be linked to the marked reduction in size observed in adipocytes.

Although the principal modifications between StD and EAARD were seen inside rpWAT, BAT was also affected. Indeed, in EAARD-fed mice, the lipid droplet dimension of BAT decreases markedly with parallel changes in UCP-1 and SIRT-3 immunostaining only.

UCP1 is a mitochondrial membrane protein consisting exclusively of BAT devoted to adaptive thermogenesis. Indeed, it can translocate protons through the inner mitochondrial membrane of brown adipocyte mitochondria, thus uncoupling respiration from ATP synthesis. As a result, UCP-1 provokes energy dissipation in the form of heat and also stimulates high levels of fatty acid oxidation [70].

SIRT-3 is a protein located in the mitochondrial matrix. Over-expression of SIRT-3 in cultured cells increases respiration and decreases the production of reactive oxygen species. Over-expression of SIRT-3 in brown adipocytes increases the expression of PGC-1α and UCP-1, indicating a role for SIRT-3 in adaptive thermogenesis [71]. Globally, our data from EAARD-fed mice show that excess EAAs is responsible for lipid decreases by increasing thermogenesis. This agrees and confirms previous results in skeletal and cardiac muscle, showing that a EAAs-rich diet improves mitochondria biogenesis, respiration and cell survival [9].

The reduced food intake observed in EAARD-fed animals mimics caloric restriction (CR). CR is a dietary intervention that delays aging and extends the period of health in various species [72]. A reduction in adiposity is a hallmark of CR, an intervention that extends longevity and delays the onset of these same age-related conditions. In addition, previous studies in adult mice showed that CR induces expression of genes involved in multiple aspects of metabolism as a consequence of the endocrine function of AT [27,73]. Indeed, adipokines and lipokines secreted from WAT impact peripheral tissue fuel utilization and the balance of energy generation from lipid or carbohydrate sources [74,75,76].

CR has also been shown to be effective in humans [77]. However, it does not seem to be a viable path in the elderly or in patients with severe chronic disease. As such, many researchers have focused on the development of CR mimetic compounds providing some of the benefits of dietary restriction without a reduction in caloric intake [78]. Unfortunately, to date, any attempts have been only partially successful in experimental models only and are not possible for humans in the immediate future.

In conclusion, our data agree and extend recent research, indicating that diet contained a balanced free EAAs mixture substitutes whole proteins, without changing caloric and macronutrient content, exert the modulation of body adiposity in mice. These effects mimic those obtained with CR and are finally effective through multiple modes by promoting increased thermogenesis [79].

### 4.1. Clinical Implications

We believe that integrating the nitrogen supply provided by diet through the supplementation of EAAs, to considerably increase the EAA/NEAA ratio, could be a pivotal intervention which could improve quality of life and life expectancy in humans too. We could speculate that the decreases in AT mass and the change in enzyme expression observed in rpWAT and BAT after EAARD in mice offer a chance to increase fatty acid oxidation in patients with metabolic syndrome by adequate nutritional interventions with supplementation in all EAAs.

In addition, excess EAAs in the diet could be a possible strategy to mimic a form of CR in humans without incurring the limitations of the traditional approach. Indeed, the CR regimen classically used experimentally (~40% reduction in caloric intake) may not be feasible for most humans, as it is accompanied by a significant reduction in quality of life and lean body mass, which may be deleterious to elderly individuals suffering from sarcopenia [80,81,82]. Therefore, dietary EAA supplementation and/or manipulation of the EAA/NEAA ratio could be a safe way to positively impact a healthy metabolism, aiding the prevention and/or treatment of chronic diseases regardless of calorie consumption.

### 4.2. Study Limitations

Our study has certain limitations that need to be discussed. We used an EAAs formulation tailored to human needs and thus presently used as nutritional supplement for humans. Furthermore, the impossibility, in our experimental settings, to carry out precise bromatological analyses of the AAs content provided by the StD, as well as to control possible variations among batches, even when provided by the same producer, is of some concern and an unexpected bias.

Another possible limitation is the exclusive application of IHC to support our results. However, as applied in a previous study [3], histopathological changes and especially the exact location of markers are possible only with immunohistochemistry and histochemistry. In contrast, molecular analysis, although much more sensitive in highlighting the presence of proteins and very often used exclusively, does require immediate freezing and homogenization of the sample. As a consequence, it does not take into account the specificities of the protein’s location such as tissue morphology and organization. This is a major limitation in the exclusive use of molecular analysis, which we believe is comparable if not superior to IHC. For these reasons, we believe that our preliminary data, even if obtained only by IHC, are worth considering forming the basis for further studies.

Unlike previous works [22], we did not separately evaluate the contribution of individual AAs with respect to body changes, as we believe that they do not represent reality, except in sporadic cases. Indeed, the ingestion of mixtures of all EAAs conveyed multiple effects deriving from their synergistic activity. Therefore, investigating the effect of individual, or a few AAs, although interesting from a doctrinal point of view, does not necessarily reflect the complexity of animal nutrition and metabolism. We could imagine the EAAs pool acting as a tango dancer. Just as the dancers move in a perfectly coordinated way, the EAAs act synergistically to best perform their function by influencing cellular metabolism. Future study will clarify the mechanism of action of stoichiometric mix of all EAAs and their role in fat metabolism further.

## Figures and Tables

**Figure 1 nutrients-14-02915-f001:**
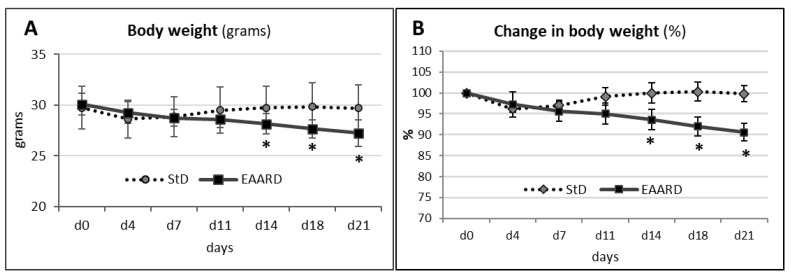
(**A**) Changes in body weight (grams, mean ± sd) during treatment. (**B**) Percentage change in body weight during treatment. EAARD-fed mice decreased body weight by approximately 10%. * *p* < 0.01.

**Figure 2 nutrients-14-02915-f002:**
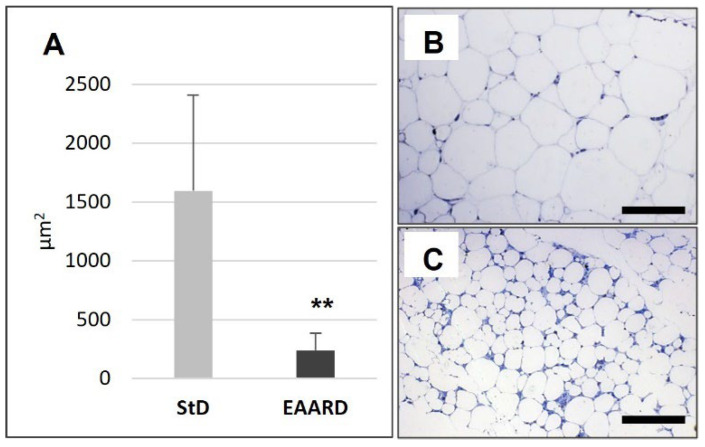
(**A**): Mean (±sd) rpWAT adipocyte area (*t* = 9.559, *p* = 0.000) according to diet. ** *p* < 0.01. (**B**,**C**): Representative images of semithin rpWAT sections toluidine blue stained from StD and EAARD, respectively. Note the dramatic size reduction in lipid droplets in EAARD mice. Original magnification 100×; scale bar 50 µm.

**Figure 3 nutrients-14-02915-f003:**
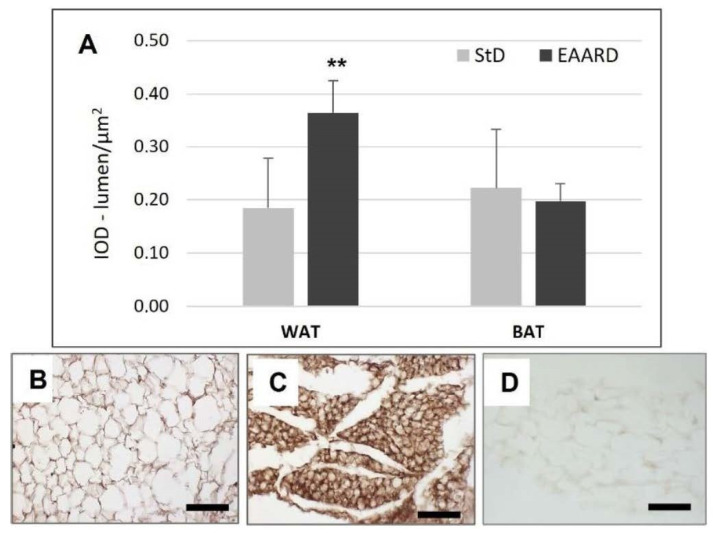
(**A**): Optical density for leptin immunostaining in rpWAT (*t* = 3.582, *p* = 0.001) and BAT (*t* = 0.998, *p* = 0.636) according to diets (mean ± sd). ** *p* < 0.01. (**B**,**C**): Representative images of rpWAT IHC in StD and EAARD-fed mice, respectively. (**D**): Negative control performed by omitting the primary antibody. Original magnification 20×; scale bar 50 µm.

**Figure 4 nutrients-14-02915-f004:**
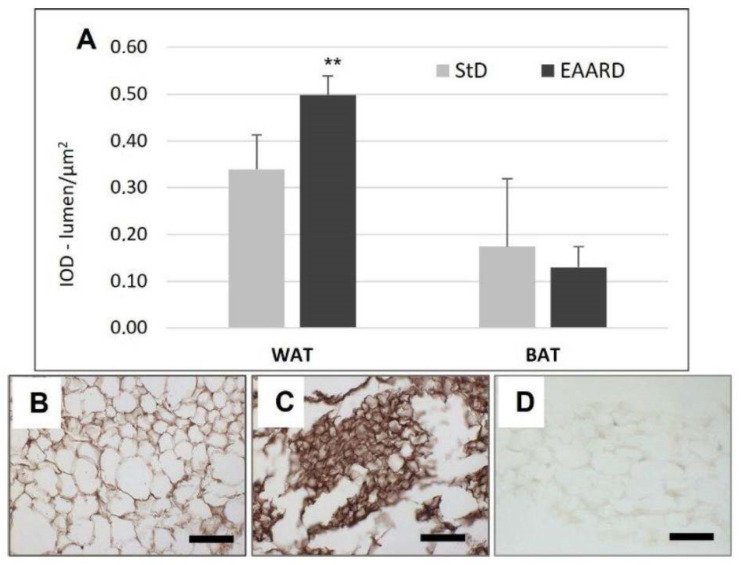
(**A**): Optical density for adiponectin immunostaining in rpWAT (*t* = 2.779, *p* = 0.01) and BAT (*t* = 2.069, *p* = 0.08) according to diets (mean ± sd). ** *p* < 0.01. (**B**,**C**): Representative images of rpWAT IHC in StD and EAARD-fed mice, respectively. (**D**): Negative control performed by omitting the primary antibody. Original magnification 20×; scale bar 50 µm.

**Figure 5 nutrients-14-02915-f005:**
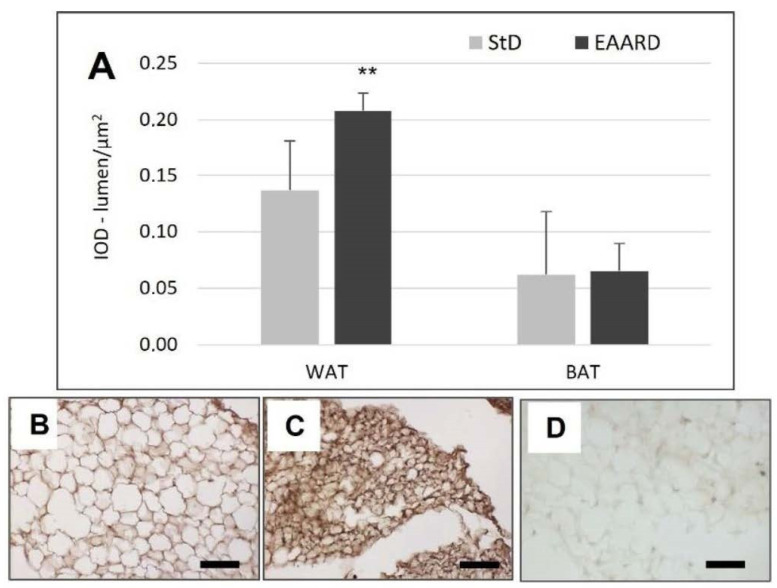
(**A**): Optical density for IL-6 immunostaining in rpWAT (*t* = 2.82, *p* = 0.01) and BAT (*t* = 0.232, *p* = 0.816) according to diets (mean ± sd). ** *p* < 0.01. (**B**,**C**): Representative images of rpWAT IHC in StD and EAARD-fed mice, respectively. (**D**): Negative control performed by omitting the primary antibody. Original magnification 20×; scale bar 50 µm.

**Figure 6 nutrients-14-02915-f006:**
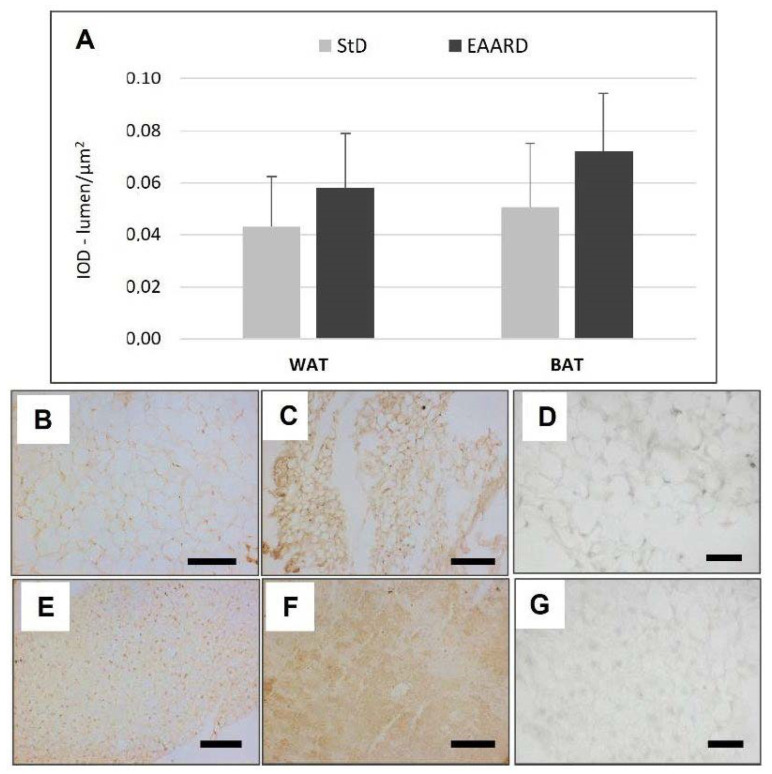
(**A**): Optical density for IL-10 immunostaining in rpWAT (*t* = 1.068, *p* = 0.571) and BAT (*t* = 1.615, *p* = 0.212) according to diet (mean ± sd). Representative images of rpWAT (**B**,**C**) and BAT (**E**,**F**) IHC in StD and EAARD-fed mice, respectively. (**D**,**G**): Negative control performed by omitting the primary antibody. Original magnification 20×; scale bar 100 µm.

**Figure 7 nutrients-14-02915-f007:**
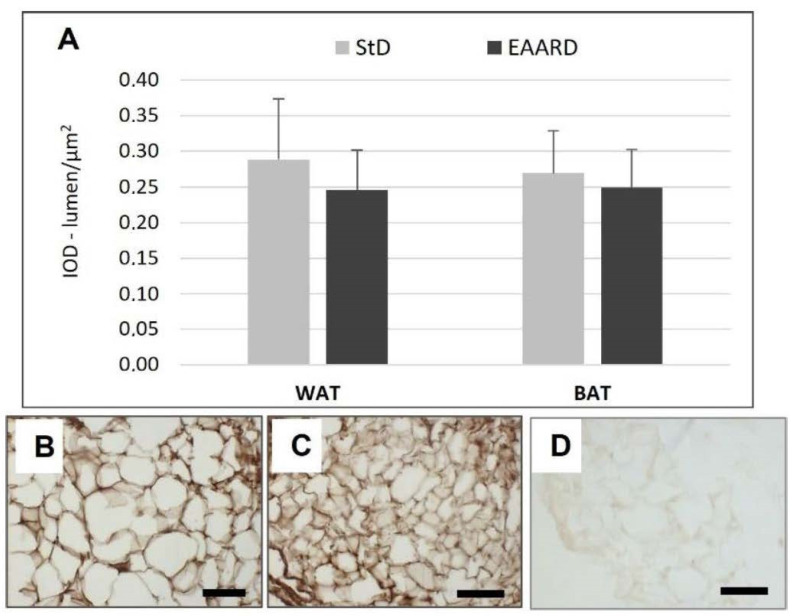
(**A**): Optical density for FAS immunostaining in rpWAT (*t* = 1.651, *p* = 0.869) and BAT (*t* = 0.081, *p* = 0.93) according to diet (mean ± sd). (**B**,**C**): Representative images of rpWAT IHC in StD and EAARD-fed mice, respectively. (**D**): Negative control performed by omitting the primary antibody. Original magnification 20×; scale bar 50 µm.

**Figure 8 nutrients-14-02915-f008:**
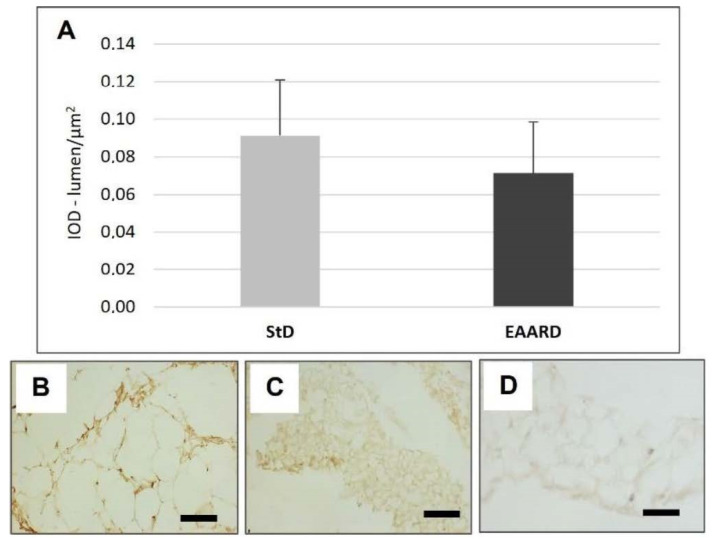
(**A**): Optical density for FADD immunostaining in rpWAT (*t* = 1.402, *p* = 0.322) according to diet (mean ± sd). (**B**,**C**): Representative images of rpWAT IHC in StD and EAARD-fed mice, respectively. (**D**): Negative control performed by omitting the primary antibody. Original magnification 20×; scale bar 50 µm.

**Figure 9 nutrients-14-02915-f009:**
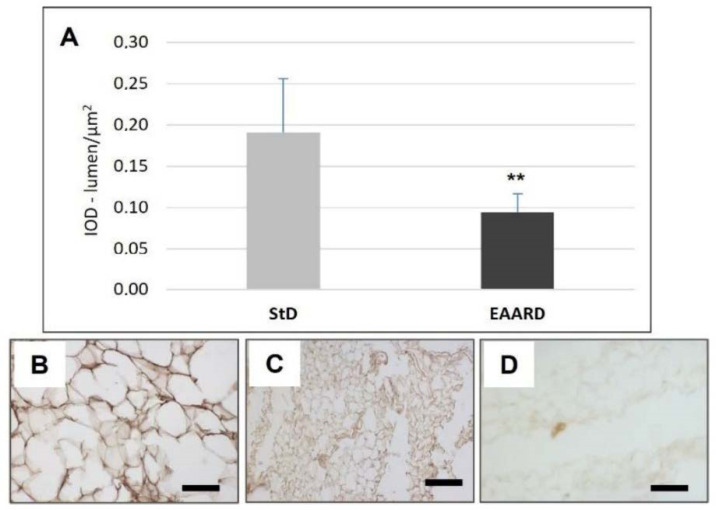
(**A**): Optical density for ATGL immunostaining in rpWAT (*t* = 3.998, *p* = 0.000) according to diets (mean ± sd). ** *p* < 0.01. (**B**,**C**): Representative images of rpWAT IHC in StD and EAARD-fed mice, respectively. (**D**): Negative control performed by omitting the primary antibody. Original magnification 20×; scale bar 50 µm.

**Figure 10 nutrients-14-02915-f010:**
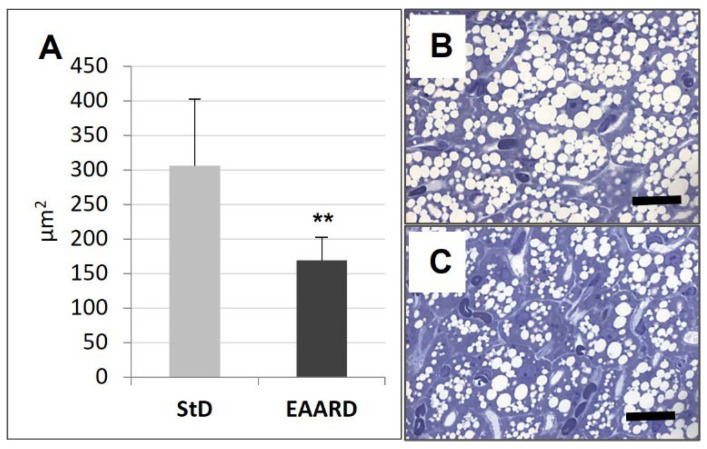
(**A**): Mean (±sd) BAT adipocyte area (*t* = 9.486, *p* = 0.000) according to diet. ** *p* < 0.01. (**B**,**C**): Representative images of semithin BAT sections toluidine blue stained from StD and EAARD, respectively. EAARD-fed mice shown the dramatic reduction in lipid droplets. Original magnification 100×; scale bar 50 µm.

**Figure 11 nutrients-14-02915-f011:**
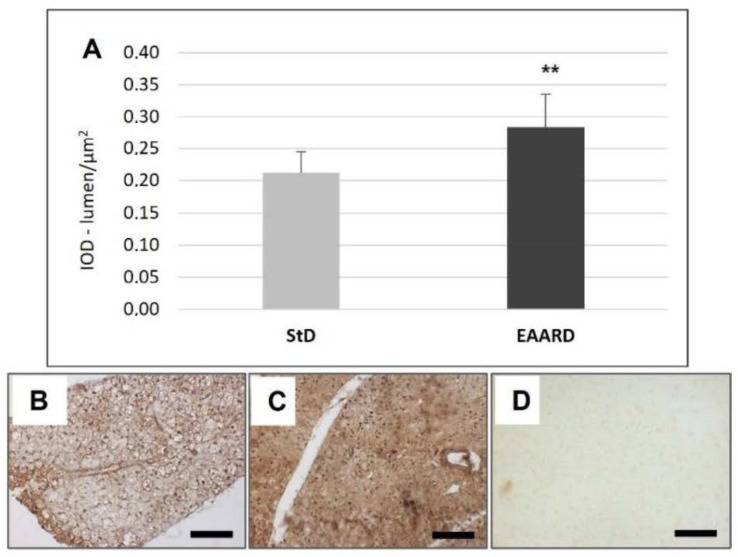
(**A**): Optical density for UCP-1 immunostaining in BAT (*t* = 3.352, *p* = 0.000) according to diet (mean ± sd). ** *p* < 0.01. (**B**,**C**): Representative images of BAT IHC in StD and EAARD-fed mice, respectively. (**D**): Negative control performed by omitting the primary antibody. Original magnification 20×; scale bar 50 µm.

**Figure 12 nutrients-14-02915-f012:**
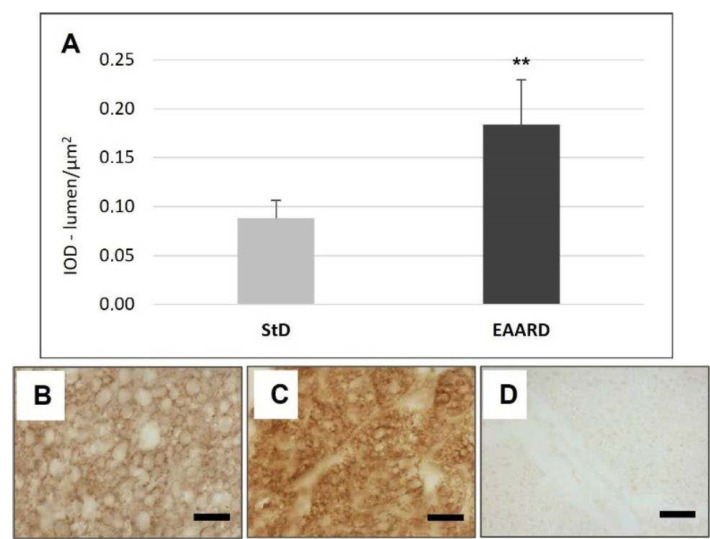
(**A**): Optical density for SIRT-3 immunostaining in BAT (*t* = 5.44, *p* = 0.000) according to diets (mean ± sd). ** *p* < 0.01. (**B**,**C**): Representative images of BAT IHC in StD and EAARD-fed mice, respectively. (**D**): Negative control performed by omitting the primary antibody. Original magnification 20×; scale bar 50 µm.

**Table 1 nutrients-14-02915-t001:** Diet composition.

	StD	EAARD
KCal/Kg	3952	3995
Carbohydrates (%)	54.61	61.76
Lipids (%)	7.5	6.12
Nitrogen (%)	21.8 °	20 *
Proteins: % of total N content	95.93	--
Free AA: % of total N content	4.07	100
EAA/NEAA (% in grams)	-	86/14
Free AA composition (%)		
l-Leucine (BCAA)	--	13.53
l-Isoleucine (BCAA)	--	9.65
l-Valine (BCAA)	--	9.65
l-Lysine	0.97	11.6
l-Threonine	--	8.7
l-Histidine	--	11.6
l-Phenylalanine	--	7.73
l-Methionine	0.45	4.35
l-Tyrosine	--	5.80
l-Tryptophan	0.28	3.38
l-Cystine	0.39	8.20
l-Cysteine	--	--
l-Alanine	--	--
l-Glycine	0.88	--
l-Arginine	1.1	--
l-Proline	--	--
l-Glutamine	--	--
l-Serine	--	2.42
l-Glutamic acid	--	--
l-Asparagine	--	--
l-Aspartic acid	--	--
Ornithine-αKG	--	2.42
N-acetyl-cysteine	--	0.97

* Nitrogen (%) from free AAs only. ° Nitrogen (%) from vegetable and animal proteins and added AA. StD = standard diet; EAARD = essential-AAs-rich diet; N = nitrogen. The continuous line represents the limit between EAAs (upside) and conditional EAAs (l-cystine and l-cysteine) and non-EAAs (beneath). BCAA = branched-chain AAs.

**Table 2 nutrients-14-02915-t002:** Mean daily food and caloric intake (mean ± sd) according to diets.

	StD	EAARD	*t*	*p*
Food (g/day)	4.43 ± 0.33	2.8 ± 0.23 *	12.142	0.00
Calories (Kcal/day)	17.51 ± 1.3	11.19 ± 0.92 *	11.906	0.00

*t*-test: * *p* < 0.05.

**Table 3 nutrients-14-02915-t003:** Organ weight (grams, mean ± sd) at the end of treatment.

	Heart	Kidney	Liver	Spleen	Triceps	rpWAT	BAT
StD	0.21 ± 0.02	0.58 ± 0.06	1.68 ± 0.2	0.15 ± 0.01	0.21 ± 0.03	0.16 ± 0.03	0.15 ± 0.02
EAARD	0.18 ± 0.02 *	0.48 ± 0.04 *	1.46 ± 0.03 *	0.12 ± 0.02 *	0.27 ± 0.02 *	0.01 ± 0.00 *	0.11 ± 0.05 *
*t*	3.286	4.140	3.088	4.431	4.968	14.137	2.489
*p*	0.002	0.000	0.004	0.000	0.000	0.000	0.025

rpWAT = retroperitoneal white adipose tissue. BAT = brown adipose tissue. *t*-test: * *p* < 0.05.

**Table 4 nutrients-14-02915-t004:** Organ weight normalized to b.w. (mean ± sd) at the end of treatment.

	Heart/bw	Kidney/bw	Liver/bw	Spleen/bw	Triceps/bw	rpWAT/bw	BAT/bw
StD	0.72 ± 0.04	2.02 ± 0.23	5.89 ± 0.92	0.54 ± 0.05	0.72 ± 0.03	0.57 ± 0.09	0.51 ± 0.04
EAARD	0.70 ± 0.02	1.84 ± 0.14	5.57 ± 0.24	0.47 ± 0.04 *	1.03 ± 0.06 *	0.04 ± 0.02 *	0.41 ± 0.17
*t*	1.309	1.982	0.962	3.315	15.263	16.389	1.958
*p*	0.381	0.095	0.672	0.002	0.000	0.000	0.100

*t*-test: * *p* < 0.05.

## Data Availability

The data presented in this study are available on request from the corresponding author.

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
