# Peer review of "Essential Amino Acids-Rich Diet Decreased Adipose Tissue Storage in Adult Mice: A Preliminary Histopathological Study"

_nutrients, 2022, doi:10.3390/nu14142915_

Round 1
Reviewer 1 Report
The authors present an interesting study on how a diet rich in essential acids affects brown and white adipose tissue storage. The manuscript is well presented, logically explained, and would be of interest to researchers in various related fields. I would recommend this manuscript for publication in 'Nutrients' journal after the following minor changes have been made (these apply more to the formatting of the document than the scientific content, which I believe to be sound):
1. The manuscript would benefit from proofreading/editing by a native English speaker or a professional editing service, as the grammar is a little poor in places.
2. Colons should be followed by a capital letter. This applies in several places of the document. In the captions of Figures 2, 3, 4, 5 and 6, there should be a capital letter to begin each sub-caption.
3. At the same time, capital letters are used in various places of the manuscript where lower case letters should be used. For example, line 74 (Nitro), line 185 (Leptin) and line 190 (Adiponectin) – this list is not exhaustive (also in lines 175–177)
4. A negative sign is better represented as an en dash rather than a standard hyphen. (e.g. line 161)
5. In Fig 1 A and B, there is inconsistency in the axis formatting.
6. Line 129: 'toluidine blue' is written in a smaller font to the rest of the document.
7. In the caption of Figure 2, the sub-caption 'D' is missing, which should refer to the negative control.
8. Abbreviations should be defined upon their first use, in both the main document and the abstract. In the abstract this applies to 'WAT' and 'BAT'. In the main document, the authors should also check the abbreviations. For example, on line 130 is the first mention of 'Immunohistochemistry' and it should therefore be defined as 'IHC' in parenthesis at this point. The authors should also consider a full list of abbreviations at the end of the document.
9. On page 7, the authors should be consistent with the spacing around the ± sign.
10. The 'L' in amino acid names, e.g. L-leucine, should be in a slightly reduced font size (Table 1).
11. Lines 172 and 173: A superscript squared term should be used.
Reviewer 2 Report
Dear Authors,
Essential amino acids rich diet decreased adipose tissue storage in adult mice: a preliminary histopathological study
is a very important study as the increase in the geriatric population is anticipated during the next 20 years. The increase in lifestyle with improvement in healthcare has significantly accelerated this phenomenon. Amino acids play a key role in metabolic processes especially maintenance of adipose tissue. Adipose tissue contributes towards maintaining the temperature of the body. Decrease in adipose tissue effects bodily temperature. This study was performed well to consider the key endogenous special tissue for this study. The evidence of histochemical studies are relevant but additional metabolic analysis would have added more conclusions and support to the hypothesis. Nevertheless, it is statistically significant and readers will learn from the experimental design. It has the potential for a clinical study as well.
The significance of branched aliphatic aminoacids was highlighted in perspective of contributing towards adipose tissue biosynthesis. A rationale for this would have been essential (lines 40 - 50). A schematic picture depicting the branched aminoacids to the biosynthetic pathway of adipose tissue would have been helpful to the readers.
Bioenergetics is connected to Krebs cycle. No discussion of this basic biochemistry cycle is discussed in this study. But evidently the paraffin cross sections of the tissue with IHC provided evidence of adipose tissue biosynthesis and aminoacid rich diet.
Author Response
Please see the attachment.

This manuscript is a resubmission of an earlier submission. The following is a list of the peer review reports and author responses from that submission.
Round 1
Reviewer 1 Report
On Essential amino acids rich diet decreased adipose tissue storage in adult mice: a preliminary histopathological study by Corsetti et al.
The authors sought to assess the “influence of balanced special diet with excess in all free EAAs on retroperitoneal WAT (rpWAT), interscapular BAT and body composition of adult mice.” The authors commissioned the synthesis of a EAA enriched diet based on parameters worked out for humans. The authors used 20 animals to test this question and used immunohistochemistry to assess the expression of some proteins in adipose tissue. The authors conclude that “diet contained a balanced free EAAs mixture substitutes whole proteins, without changing caloric and macronutrient content, exert anti-obesity effects, with modulation of body adiposity in mice.”
This reviewer is excited to continue to see work being done on the effects of amino acids on metabolism. However, the author’s conclusions are not supported by their data and a significant amount of work needs to be done in order to be able to draw an accurate conclusion from this experimental design. Below are a list of major issues identified.
Major Comments.
- The authors state that this study was done in 20 mice. This indicates that the findings have not been replicated which is a major concern for rigor and reproducibility.
- the authors acknowledge that the diet composition was based on human studies. This is problematic due to the multiple lines of research indicating that AAs when in high levels, can exert toxic effects in mice.
- The authors claim that the enriched EAA diet caused significant weight loss. This is inaccurate as we have to take into consideration the fact that mice in that group were eating about 40% less calories than control mice. This is a big red flag and indicates a potential toxic effect of the diet which must be ruled out. In addition, this likely resulted in unrestrained lipolysis in adipose tissue leading to the smaller adipocytes but more of a survival effect rather than a physiological one. It is unclear why the authors chose to only focus on rpWAT and BAT which are the least expandable adipose depots in a mouse under standard chow conditions? Why not the other WATs?
- the majority of the conclusions in this manuscript are based on histological assessments which is just not enough. In addition, the histology quality is subpar with adipocytes clearly broken in the sectioning process and no clear delineation of the cells. While the authors state that they had negative controls, none are shown in the main figures as they should. The majority of the histology looks to be non-specific binding/effects/overdeveloping of the DAB.
- Methods lack depth of detail. It is unclear how the tissues were processed and sectioned e.g. what thickness? For paraffin embedded samples, the quality is not up to par. How long were samples fixed for? How were they treated after etc. How was food intake assessed? Group? Singly house? The authors indicate that the “animas fed with EAARD were not suffering according to veterinary advice” but were there signs of lethargy or sickness? It is highly surprising to this reviewer that a mouse eating <40% of the normal kcal/day will be healthy.
- Statistical analyses are not appropriate. The authors indicate that “Statistical analysis was performed by one- way ANOVA followed by two-sample unpaired t-Test”. The author only had 2 groups to compare (SD vs EAARD) and thus an ANOVA is not called for which is used for >3 groups. If they are referring to the body weight curves then that needs to be clarify and all comparisons expressed. All other comparisons need to be assessed appropriately.
Overall, this reviewer thinks the overall concept is of interest and can provide much needed insight in the field of the effects of amino acids on metabolism but the current study does not meet the expectations to make any conclusions regarding the effects of AA on adiposity.
Reviewer 2 Report
This study focus on the effects of essential amino acids rich diet on adipose tissue storage in adult mice. It is an intersting work to provide food choice for people. Besides, I have some concerns about this study.
- In the Figure 1, did the body weight in these two group show significant changes?
- In the Figure 1B, error bar was needed.
- In the Table 4, the ratio of spleen/body weight was decreased in EAARD group. Did the spleen function of EAARD mice impair? Did the drink water and urinary output of these mice change?
- In the results, why did choose these factors to estimilate the effects of EAARD? More details are needed as it is hard to follow.
- Leptin is an important hormol to promote satiefy, not suppress.
- As an proinflammatory factor, IL-6 is able to induce oxidative stress. However, what was the level of IL-6 after EAARD treatment? As all known, physiological or pathological enhancement of IL-6 leads to different outcomes. It is necessary to compare the IL-6 level in normal and obese status.
- It is elusive of the logic in the discussion. More unrelated information was absorbed.
- This study aims to declear that EAARD could be a pivotal intervention that could improve the life expectancy of malnourished people. However, the authors used healthy mice to perform all expreiments, which is not consistent with the conclusion.
- Standard English editing are needed.
Reviewer 3 Report
Journal: Nutrients
Title: Essential amino acids rich diet decreased adipose tissue storage in adult mice: a preliminary histopathological study.
The authors describe the effect of the influence of a special diet containing all EAAs on retroperitoneal WAT (rpWAT) and interscapular BAT of mice. They analysed histochemically several parameters as well as the weight gain of the mice. They showed a clear beneficial effect of the EAA diet.
However, there are some points, which need improvement or clarifying.
- English has to be improved. Several phrases are difficult to understand or sound strange and there are some typing errors. Please give the manuscript to a native English-speaking person to revise it.
- Table 1 is not clear: it is written "The EAA-Rich Diet (EAARD) contained a mixture of EAAs (84%) and NEAAs (16%)" and " EAA/NEAA (% in grams) 84/16" but the sum of
EAA (Leucine to Tryptophan) is "composition (%)" 92.53,
and the sum of
NEAA (Alanine to acetyl-cysteine) 4.67
how is an 84 to 16 ratio achieved? - Figure 10. shows BAT adipocyte's area, but in the legend stands "representative images of semithin rpWAT sections toluidine blue stained" please correct.
- Line 293: is written "Up-regulation of anti-inflammatory IL-10 secretion is induced by adiponectin by increased tissue inhibitor of metalloproteinase-1 in human macrophages [55]." However, according to the title of ref 55, (Adiponectin specifically increased tissue inhibitor of metalloproteinase-1 through interleukin-10 expression in human macrophages.) The sequence is adiponectin => IL10 => metalloproteinase-1, please correct.
- Line 364: it is written that an increase of the EAA/NEAA ratio to at least >1, could be a pivotal intervention that could improve the life expectancy. But in the introduction is written " Dietary animal and vegetable proteins are the unique source of EAAs, ... Unfortunately, any dietary proteins contain an excess of NEAAs (EAA/NEAA ≤0.9 at best, with a mean % of about 45/55 respectively) ... the food variety hardly allows the constant absence of one or more EAAs."
If a normal diet contains a ratio of around 0.9 and in the experiments a ratio of at least 5 was used to obtain the results, why should a ratio of >1 ( means e.g. 1.1 or 1.3) already be beneficial?